# Genomic Characterization of Marine *Staphylococcus shinii* Strain SC-M1C: Potential Genetic Adaptations and Ecological Role

**DOI:** 10.3390/microorganisms13081866

**Published:** 2025-08-09

**Authors:** Manar El Samak, Hasnaa Lotfy, Abdelrahman M. Sedeek, Yehia S. Mohamed, Samar M. Solyman

**Affiliations:** 1Department of Microbiology & Immunology, Faculty of Pharmacy, Suez Canal University, Ismailia 41522, Egypt; 2Department of Microbiology & Immunology, Faculty of Pharmacy, Sinai University—Elkantara Branches, Ismailia 41522, Egypt; hasnaa.kamel@su.edu.eg; 3Department of Microbiology & Immunology, Faculty of Pharmacy, Galala University, New Galala City 43511, Egypt; abdelrahman.sedek@gu.edu.eg; 4Department of Pathological Sciences, College of Medicine, Ajman University, Ajman P.O. Box 346, United Arab Emirates; 5Department of Microbiology and Immunology, Faculty of Pharmacy (Boys), Al-Azhar University, Cairo 11562, Egypt

**Keywords:** Red Sea, sponge-associated bacteria, microbial genomics, *Staphylococcus shinii*

## Abstract

*Staphylococcus shinii* (*S. shinii*) is a coagulase-negative species primarily associated with the degradation of organic matter, contributing to nutrient cycling in natural environments. This species has been mainly studied in clinical and terrestrial contexts, with no previous reports of its presence in marine environments. In this study, we report the first isolation of *S. shinii* from a marine habitat. The strain SC-M1C was isolated from the Red Sea sponge *Negombata magnifica*. Whole-genome sequencing confirmed its taxonomic identity as *S. shinii*. The genome uncovers potential adaptive characteristics that facilitate survival in marine ecosystems, comprising genes associated with osmoregulation, nutrient acquisition, stress response, and resistance to heavy metals. Moreover, multiple genomic islands and plasmids were identified, suggesting a potential role in horizontal gene transfer and environmental adaptability. The presence of biosynthetic gene clusters linked to non-ribosomal peptides, siderophores, and terpene production indicates potential for biochemical versatility beyond traditional metabolic expectations. This study presents the first genomic insights into *S. shinii* in a marine context, highlighting its ecological significance and adaptive mechanisms in a high-salinity environment. These findings expand our understanding of staphylococcal ecology beyond terrestrial and clinical origins and provide a foundation for exploring the role of *S. shinii* in marine microbial interactions and environmental resilience.

## 1. Introduction

Microbes are considered key players in maintaining the health of marine hosts, forming complex symbiotic relationships with invertebrates such as sponges, corals, and mollusks. These associations often confer benefits ranging from defense against pathogens to the provision of essential nutrients [1,2]. With the increasing exploration of extreme marine environments, scientists are uncovering novel microbial lineages with biotechnological and medical potential, in addition to adaptive mechanisms that challenge traditional views of microbial physiology and evolution [3,4].

The Red Sea represents a unique marine ecosystem characterized by high salinity, elevated temperatures, and distinct seasonal variations. These extreme conditions foster a rich diversity of microbial life, including complex symbiotic communities associated with marine invertebrates [5,6,7]. Sponge-associated microorganisms play crucial roles in nutrient cycling, host defense, and environmental resilience, making them key players in marine microbial ecology [8]. Despite their ecological importance, many bacterial taxa within these sponge microbiomes remain underexplored, particularly non-pathogenic genera traditionally associated with terrestrial or clinical environments.

Unlike dominant marine species like *Vibrio*, *Pseudomonas*, and *Alteromonas*, which have been the major focus in research studies, the *Staphylococcus* species has received less attention in marine settings, despite their emerging ecological and biotechnological significance [9,10]. *Staphylococcus* is a low-GC Gram-positive bacterial group and is commonly found in diverse habitats ranging from soil and water to human and animal hosts [11]. While species like *Staphylococcus aureus* are highly pathogenic and cause various infections, many others, such as *S. epidermidis*, *S. lugdunensis*, *S. saprophyticus*, and *S. haemolyticus*, have a more restricted pathogenic potential or are primarily commensal. In addition, various *Staphylococcus* species, especially those isolated from environmental sources or utilized in food production (e.g., *S. carnosus*, *S. xylosus*, and *S. equorum*), are typically regarded as non-pathogenic [12,13,14]. In the context of marine Staphylococci, various *Staphylococcus* species have been previously isolated from different sponge samples under various culture conditions [15,16,17].

*S. shinii* has recently been described and validated as a novel species [18]. Meanwhile, 25 genomes of this species have been listed in the National Center for Biotechnology Information (NCBI) genome database (accessed June 2025). The NCBI-listed strains were found to be isolated from different sources, such as Korean fermented bean paste, cattle parasites, raw cow’s milk, river water, bovine mastitis collection, and a human nasal swab [19,20]. Unlike its pathogenic relatives, *S. shinii* is considered a part of microbial communities associated with organic degradation.

Currently, whole-genome sequencing has become an essential tool for understanding bacterial biology, offering insights into mechanisms of adaptation, virulence, antimicrobial resistance, and horizontal gene transfer that allow bacteria to survive under extreme conditions, interact with their environment, and evolve in response to selective pressures [21,22,23]. However, genomic studies of *S. shinii* have primarily focused on terrestrial or clinical isolates, leaving its genetic traits and potential ecological roles in marine environments unexplored.

This study aims to elucidate the genetic adaptations and ecological functions of the *S. shinii* strain SC-M1C, a marine bacterium isolated from a sponge, with a focus on its potential roles in marine ecosystems. To address the current knowledge gap, we present a comprehensive genome-wide analysis of this marine isolate. Our findings expand the understanding of Staphylococcus ecology beyond terrestrial and clinical settings, contributing to the broader exploration of microbial adaptation in extreme marine environments. This work also underscores the importance of further investigating the ecological roles of overlooked bacterial taxa within sponge-associated communities.

## 2. Materials and Methods

### 2.1. Sample Collection and Strain Isolation

The strain SC-M1C was isolated from the Red Sea sponge *Negombata magnifica* (Figure 1a). The sponge was collected on 10 September 2020, from a sandy bottom at a depth of 6 m in Naama Bay, Sharm El-Sheikh, Sinai, Egypt (Figure 1b). The sponge identification was undertaken morphologically, and the bioactivity of the sponge’s microbiome has been previously reported in our laboratory [6,24]. Sponge tissue samples, kept at ambient seawater temperature, were transported to the lab in sterile bags within less than 8 h of collection. Approximately 1 g of the sample was then aseptically removed, rinsed with filtered and autoclaved natural seawater (NSW), and homogenized by vortexing in 10 mL of sterilized NSW. Tenfold serial dilutions (10^−1^ to 10^−6^) were prepared in sterilized NSW, and 100 µL aliquots were plated on Reasoner’s 2A agar (R2A) (Difco^TM^, Detroit, MI, USA) supplemented with 2% NaCl using a sterile spreader to disperse around 100 μL of each dilution onto R2A plates. The plates were then incubated at 30 °C. A total of twenty isolates were recovered, 19 of which were reported to have antimicrobial activity [6]. SC-M1C strain showed full growth and characteristic colony morphology within 24 h. Following isolation, the strain was further purified by repetitive sub-culturing on marine agar, and preserved as a glycerol stock at −80 °C.

### 2.2. DNA Extraction and Whole-Genome Sequencing

The bacterial isolate was grown overnight in R2A broth at 30 °C while shaking at 150 rpm until the log phase of growth was reached. The grown culture was centrifuged at 10,000 rpm for 10 min to pellet the bacterial cells, and the supernatant was discarded. DNA was extracted from the pelleted bacteria using the DNA extraction protocol of the FavorPrep™ Tissue Genomic DNA Extraction Mini Kit (Favorgen Biotech Corp., Hsinchu City, Taiwan) for Gram-positive bacteria. DNA quantity and purity were assessed with a NanoDrop^TM^ 1000 Spectrophotometer V3.7 (Thermo Fisher Scientific Inc., Wilmington, DE, USA). The genomic DNA was then shipped to Admera Health, LLC in the USA for library preparation using KAPA Hyper Prep PCR Free and paired-end 150 bp sequencing on a HiSeq 6000 instrument (Illumina, San Diego, CA, USA).

### 2.3. Reads Preprocessing and Assembly

For quality control, reads were filtered using Trimmomatic version 0.39 [25]. This process involved clipping Illumina adaptors, applying a sliding window trim of at least 4 bases, and setting a minimum average quality score of 20. Subsequently, these filtered reads were assembled with the Unicycler version 0.4.8 assembler via the BV-BRC server [26].

### 2.4. Strain Typing and Phylogeny

A phylogenomic-based genome analysis between the genome of the isolate SC-M1C and the top-related strain types was conducted on the Type Strain Genome Server (TYGS) [24]. To ensure the taxonomic affiliations of both isolates, the average nucleotide identity (ANI) and DNA–DNA hybridization (DDH) were calculated using the JSpeciesWS (web server) [27], employing both BLAST-based alignment (ANIb) and MUMmer-based alignment (ANIm) with default parameters and a genome-to-genome distance calculator [28], respectively.

### 2.5. Reference-Guided Scaffolding and SC-M1C Genome Annotation

RagTag version 2.1.0 was used for the reference-guided scaffolding using the default parameters [29]. Rapid annotations using Subsystems Technology (RAST) [30] and Prokka [31] were used to annotate the SC-M1C genome. The Comprehensive Antibiotic Resistance Database (CARD), the National Database of Antibiotic Resistant Organisms (NDARO), and PATRIC [32] were used to annotate antibiotic resistance genes [33]. The IslandViewer 4 server was used to locate the genomic islands (GIs) [34]. To investigate the main metabolic processes in the strain SC-M1C, the KEGG was used for KO annotations and reconstruction of metabolic pathways [35]. The Cluster of Orthologous Genes (COG) database was used to classify SC-M1C protein sequences into the different COG functional categories [36]. The PlasmidFinder version 2.1 was used to search for plasmid replicons within the genome [37]. The BGCs accountable for the production of secondary metabolites and their similarities to known metabolites were detected by AntiSMASH bacterial version 6.0 [38]. The VFDB server was used to identify the virulence factor-related genes [39].

## 3. Results

### 3.1. The Isolated Strain’s Phenotypic Characteristics

On nutrient agar, the isolate formed smooth, circular, convex, and opaque colonies with entire margins and a pale-yellow pigmentation, typically measuring 1–2 mm in diameter after 24 h of incubation at 30 °C (Figure 2a). Microscopic examination of the isolated strain revealed Gram-positive, spherical cocci arranged in irregular clusters. The microorganism was non-spore-forming, non-motile, and catalase-positive, consistent with the general characteristics of the *Staphylococcus* genus (Figure 2b).

### 3.2. Genome Characteristics and Strain Typing of Isolate SC-M1C

The post-filtering read statistics resulted in 18,112,185 sequences and a total length of 2.7 Gbp (average sequence length 151 bp per sequence). No sequences were flagged as poor quality. The genome sequencing yielded 36 contigs with an N_50_ value of 522,659 bp, a GC content of 32.34%, and a total genome length of 3,223,632 bp. Strain typing via the Type Strain Genome Server (TYGS) identified the isolated strain as *S. shinii*, a low GC-content, Gram-positive bacterium within the Bacillota (formerly Firmicutes), class Bacilli, and order Bacillales. To confirm the taxonomic positions of the isolate, digital DNA–DNA hybridization (dDDH), and average nucleotide identity (ANI) comparisons were performed with their closest type strains, as the type strain “*S. shinii* K22-5M” exhibited the closest relationship, with dDDH, ANIm, and ΔCG% values of 94.8%, 99.40% and 0.15%, respectively (Table 1). These values confirm the taxonomic position of the isolated strain. A phylogenomic tree depicting the evolutionary relationships between both strains and their closely related type strains is presented in Figure 3.

### 3.3. S. shinii Strain SC-M1C Genome Annotation and Genome Mapping

The Similar Genome Finder tool identified *Staphylococcus pseudoxylosus* strain 14AME19 (GenBank: CP068712.1) as the closest complete genome available on the public database to the strain SC-M1C, with a distance of 0.00730426, making it suitable for reference-guided scaffolding. Using RagTag, the genome of the strain SC-M1C was scaffolded into eight contigs, with the largest spanning 3,052,874 bp and the smallest 5491 bp, resulting in a coverage of 98.7% of the reference genome.

The RAST and Prokka pipelines were used to annotate the genome of *S. shinii* strain SC-M1C, as RAST returned more annotations with functional assignments than Prokka (Table 2). The genome map of the *S. shinii* strain SC-M1C is illustrated in Figure 4.

Approximately 80.89% of the coding genes within the genome of *S. shinii* strain SC-M1C were assigned to the COG functional category; with 5.97% related to inorganic ion transport and metabolism; 3.01% related to signal transduction mechanisms; 5.66% related to cell wall, membrane, and envelope biogenesis; and 1.81% related to secondary metabolites biosynthesis, transport, and catabolism (Figure 5).

### 3.4. Antimicrobial Resistance and Horizontal Gene Transfer

While the scaffolded contig 1 represents the bacterial chromosome, contigs from 2 to 8 contain plasmid-related genes. Plasmid analysis using PlasmidFinder version 2.0 identified specific plasmid replicons in these contigs. A replicon belonging to the Inc18 family was detected, with a 93.04% sequence identity to the rep16 replicon (GenBank: CP000737), located on contig number 4 at positions 3121–3637. Additionally, two replicons from the RepA-N family were identified: rep20, exhibiting 90.95% identity (GenBank: HE616681) and found on contig number 3 at positions 6541–7479, and rep24c, showing 97.38% identity (GenBank: FR687301) and located on contig number 6 at positions 422–1414. These findings indicate the presence of plasmids associated with the Inc18 and RepA-N families, which are commonly linked to functions such as antibiotic resistance, conjugative transfer, and stress response, highlighting their potential role in adaptive mechanisms within the genome. Furthermore, a total of 11 virulence factors were annotated based on the virulence factor database (VFDB) within the genome of SC-M1C (Table 3).

On the other hand, a total of 12 antibiotic resistance genes were identified in the CARD database, along with 2 in NDARO and 42 in PATRIC (Table 4).

IslandViewer 4, using the complete genome of *S. shinii* strain 14AME19 (GenBank: CP068712) as a reference, identified 13 GIs, ranging in size from 5152 bp to 83,760 bp, within the genome of *S. shinii* strain SC-M1C (Figure 6). Notably, the presence of transposases, phage-related genes, integrases, and mobilization proteins within these islands underscores their potential for horizontal gene transfer, facilitating genetic adaptability and ecological fitness in dynamic marine environments (Appendix A).

### 3.5. Reconstruction of S. shinii Strain SC-M1C Metabolic Pathways

To investigate the secondary metabolite potential of the studied genome, biosynthetic gene clusters (BGCs) were predicted and analyzed using the AntiSMASH server. Twelve BGCs were identified, representing diverse classes, including nonribosomal peptide synthetases (NRPS), type III polyketide synthases (T3PKS), lanthipeptides, siderophores, and terpene-related clusters. Two of the BGCs showed 100% identity with known clusters, namely staphylopine (opine-like metallophore) and staphyloferrin A (siderophore), suggesting conserved pathways for metal acquisition. The remaining clusters displayed various biosynthetic types but lacked direct matches to characterized metabolites, indicating the potential for novel compound production. The details of the identified BGCs are summarized in Table 5.

For reconstruction of the metabolic pathways in the strain SC-M1C, the KEGG was used for annotating the CDS-predicted protein products. From 3077 CDs within the genome of *S. shinii* strain SC-M1C, there were 1630 protein products (53.0%) were annotated on the KEGG database. The annotated proteins were incorporated into 214 KEGG pathways. There were 59 complete KEGG modules, with multiple of them being functionally and ecologically significant, such as the betaine module (M00555), the formaldehyde assimilation module (M00345), and menaquinone biosynthesis module (M00116).

To adapt to the dynamic and often extreme conditions of marine ecosystems, microorganisms have evolved a range of genomic and metabolomic strategies that collectively support ecological adaptation. Key among these is osmoregulation and salt tolerance, biofilm formation and surface adhesion, nutrient acquisition, stress response, and symbiosis.

#### 3.5.1. Osmoregulation and Salt Tolerance

To thrive in marine environments characterized by fluctuating and often high salinity, microorganisms must tightly regulate intracellular osmotic pressure. The genome of *S. shinii* strain SC-M1C reveals a powerful osmo-regulatory system composed of multiple Na^+^/H^+^ antiporters and compatible solute transporters that collectively maintain ionic balance and cellular homeostasis (Table 6).

The *mnh*A1–G1 genes encoding the multi-subunit Na^+^/H^+^ antiporter were likely organized in a single operon-like cluster, suggesting coordinated expression. The *nhaC* gene was present as a single-copy gene. In contrast, five homologous copies of *opuD* were identified across the genome, indicating gene duplication and possible functional diversification in compatible solute uptake.

#### 3.5.2. Nutrient Acquisition

Marine environments are often oligotrophic, demanding efficient nutrient scavenging strategies. The genome encodes transporters and metabolic enzymes that facilitate the uptake and assimilation of essential but limited nutrients, including nitrogen, phosphate, methionine, thiamine, and iron (Table 7).

Moreover, *S. shinii* strain SC-M1C possesses several specialized metabolic pathways dedicated to efficient nutrient acquisition. For instance, the detection of the complete staphyloferrin A module (M00876) reveals a capacity for high-affinity iron scavenging through siderophore production, a critical advantage in iron-depleted seawater (Figure 7).

Similarly, the presence of the staphylopine biosynthesis module (M00949) indicates an ability to acquire essential trace metals such as nickel and cobalt via metallophore-mediated transport (Figure 8).

#### 3.5.3. Stress Response and Resistance

Exposure to environmental stressors such as UV radiation, oxidative agents, and metal ions is common in marine habitats. Several genes encode stress defense mechanisms, including oxidative damage repair, redox balance maintenance, and heavy metal detoxification (Table 8).

#### 3.5.4. Aromatic Hydrocarbon Degradation

The genome of *S. shinii* SC-M1C harbors genes implicated in the degradation of aromatic hydrocarbons (Table 9). These genes suggest a potential capacity for detoxifying aromatic and quinone-based compounds, which may contribute to the strain’s adaptability within the chemically complex sponge microenvironment.

#### 3.5.5. Two-Component Systems in *S. shinii* Strain SC-M1C

Analysis of the *S. shinii* SC-M1C genome revealed the presence of a diverse array of two-component systems (TCSs), comprising a total of 30 TCS-related proteins. This included 14 histidine kinases (12 classified as classic histidine kinases and 2 possible incomplete kinases), 15 response regulators, and 1 phosphotransferase protein.

## 4. Discussion

Marine Firmicutes play a crucial role in marine ecosystems, contributing to nutrient cycling and the production of diverse bioactive compounds with potential pharmaceutical applications [58,59]. While genera such as Bacillus have been extensively studied, marine Staphylococcus species—particularly those associated with sponges—remain largely underexplored. To our knowledge, this study represents the first comprehensive genome-wide analysis of an *S. shinii* strain from the marine environment. The sponge-associated marine bacterium *S. shinii* strain SC-M1C represents a valuable opportunity to examine the genomic features related to the resilience to challenging conditions in the marine environment. While the genome of *S. shinii* strain SC-M1C was about 3 Mbp in length, previous studies of marine *Staphylococci* genomic analysis recorded a genome size of around 2.5 Mbp [60,61], which may suggest a more complex metabolic repertoire of the *S. shinii* strain, making it interesting for study.

Genomic islands (GIs) are type of mobile genetic elements that integrate within bacterial genomes. These regions often exhibit distinct nucleotide compositions, including variations in GC content and codon usage, and are frequently flanked by mobility elements such as direct repeats, insertion sequences, or integrase genes, which facilitate their integration into the host genome [62]. GIs typically encode clusters of functionally related genes that confer adaptive advantages, such as virulence factors, antibiotic resistance, metabolic versatility, or symbiotic capabilities [63,64]. GIs encode proteins involved in processes such as DNA uptake, cell morphology regulation, stress tolerance, nutrient transport, and defense mechanisms against phages and environmental stresses.

GIs I, II, and III carry genes associated with DNA uptake, cell surface modification, and stress response, suggesting roles in niche adaptation and survival under marine stressors (e.g., high salt, limited nutrients, or oxidative conditions) [65,66,67,68]. Additionally, several islands exhibit specialized potential functions that enhance the metabolic versatility and resilience of the *S. shinii* strain SC-M1C. For example, GI X encodes arsenical/cadmium resistance operons and two-component systems linked to heavy metal resistance, heavy metal detoxification, and stress signaling. The largest island, GI XIII, encompasses a variety of genes implicated in cell wall remodeling, protein folding, DNA repair, and stress response, highlighting its importance in maintaining genomic integrity and cellular homeostasis. Collectively, these GIs reflect a mosaic of horizontally acquired traits that likely enable *S. shinii* to thrive in its marine habitat by conferring adaptive advantages through enhanced metabolic capabilities, structural adaptations, and robust defense mechanisms against biotic and abiotic challenges.

The AntiSMASH-based screening uncovered an array of cryptic BGCs. With the exception of the staphylopine and staphyloferrin A BGCs, the remaining clusters displayed various biosynthetic types but lacked direct matches to characterized metabolites, indicating the potential for novel compound production. Staphylopine and staphyloferrin A are metallophores produced by *Staphylococcus* species that support survival in metal-limited settings. Staphyloferrin A is a citrate-based siderophore that specifically chelates ferric-iron for import via the HtsABC transporter system [69,70]. Staphylopine, on the other hand, is a broad-spectrum metallophore primarily involved in zinc (and also nickel/cobalt) acquisition under conditions of host nutritional immunity, imported via the CntABC system [71,72,73,74]. While most research focuses on host–pathogen interactions, marine microorganisms commonly produce siderophores as an adaptive strategy to acquire iron in extremely iron-poor seawater [71].

Moreover, the presence of a complete KEGG module of betaine (M00555) suggests a specialized capacity for osmoregulation. Betaine functions as a potent osmoprotectant, stabilizing cellular structures and maintaining turgor pressure under high-salinity conditions [75]. This pathway likely confers a significant survival advantage, enabling SC-M1C to withstand the osmotic stresses characteristic of marine ecosystems. The ability to synthesize betaine not only supports resilience during salinity fluctuations but may also facilitate colonization of diverse marine niches.

Additionally, the formaldehyde assimilation module (M00345) equips SC-M1C to utilize one-carbon compounds as alternative carbon sources, likely exploiting metabolic byproducts produced by surrounding microbial communities [76]. These specialized systems underscore the strain’s metabolic flexibility and its potential to dominate in competitive, nutrient-limited marine ecosystems. Together, these adaptive pathways underscore SC-M1C’s capacity to endure dynamic and challenging environmental stresses.

To withstand the osmotic pressure characteristic of marine environments, *S. shinii* strain SC-M1C possesses a genetic repertoire associated with osmoregulation and salt tolerance. For instance, the *mnh* gene cluster, widely recognized for its role in maintaining ionic balance and conferring tolerance to high salinity and alkaline conditions in Gram-positive bacteria such as *Staphylococcus aureus* and Bacillus species, is present in the SC-M1C genome as the mnhA1–G1 operon. Its presence likely contributes to the halotolerant phenotype of the strain [77]. Furthermore, the genome harbors five homologous copies of opuD, which encode glycine betaine transporters essential for osmoprotection under salt stress [78]. This transporter system is evolutionarily conserved and functionally active in a wide range of bacterial species, including marine-associated taxa.

In marine ecosystems, the degradation of aromatic hydrocarbons, especially polycyclic aromatic hydrocarbons (PAHs), is a critical microbial process involving numerous genes encoding hydrocarbon-degrading enzymes. The SC-M1C genome contains the *catE* gene, a key component of the meta-cleavage pathway involved in aromatic compound degradation [79]. The *catE* gene is commonly found in a wide range of hydrocarbon-degrading bacteria, including species from the genera *Pseudomonas*, *Bacillus*, and *Rhodococcus*, and serves as a genetic marker for monitoring aromatic hydrocarbon degradation in polluted environments [80,81]. Moreover, the genome of SC-M1C also carries the genes *mhqR*, *mhqA*, *mhqO*, and *mhqD*, which are primarily associated with the degradation of hydroquinone and other phenolic compounds. These intermediates are often generated during the breakdown of lignin-derived or other complex aromatic substrates [82].

Additionally, *S. shinii* has a diverse collection of TCS-related genes. TCSs are crucial for bacterial dynamism and resilience to various environmental stresses. These systems enable bacteria to sense changes in their surrounding ecosystem and respond accordingly, ensuring their ability to adapt to diverse conditions [83,84]. The overall repertoire of TCSs in SC-M1C highlights its capacity to adapt to diverse environmental conditions typically encountered in marine ecosystems, such as nutrient fluctuations, osmotic stress, and interactions with other microbial species.

## 5. Conclusions

*S. shinii* was isolated from a Red Sea sponge in a marine environment. Key adaptations for high-salinity survival are highlighted in the genome, such as genes related to heavy metal tolerance, osmoregulation, nutrition uptake, and stress response. The presence of various genomic islands and plasmids suggests robust horizontal gene transfer mechanisms and improved environmental adaptability. Furthermore, the identification of biosynthetic gene clusters for siderophores (staphyloferrin A) and metallophores (staphylopine) points to a versatile metabolism for metal acquisition. Future research should prioritize transcriptomic validation to confirm gene expression under specific marine stresses and conduct bioactivity screening of novel compounds derived from the identified biosynthetic pathways.

## Figures and Tables

**Figure 1 microorganisms-13-01866-f001:**
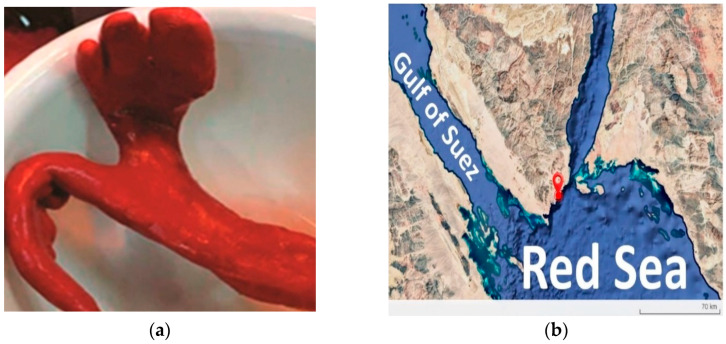
(**a**) *Negombata magnifica* sample collected from the Red Sea and (**b**) map showing the collection site of *Negombata magnifica* in the Red Sea.

**Figure 2 microorganisms-13-01866-f002:**
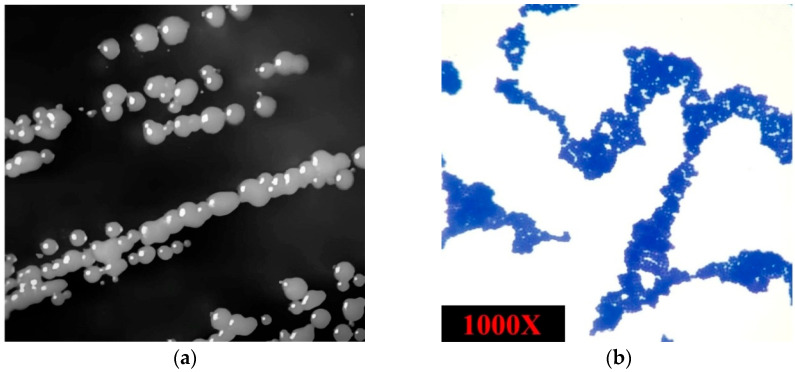
(**a**) Colony morphology of *S. shinii* strain SC-M1C on nutrient agar after 24 h and (**b**) *S. shinii* strain SC-M1C under microscope using 1000× magnification power.

**Figure 3 microorganisms-13-01866-f003:**
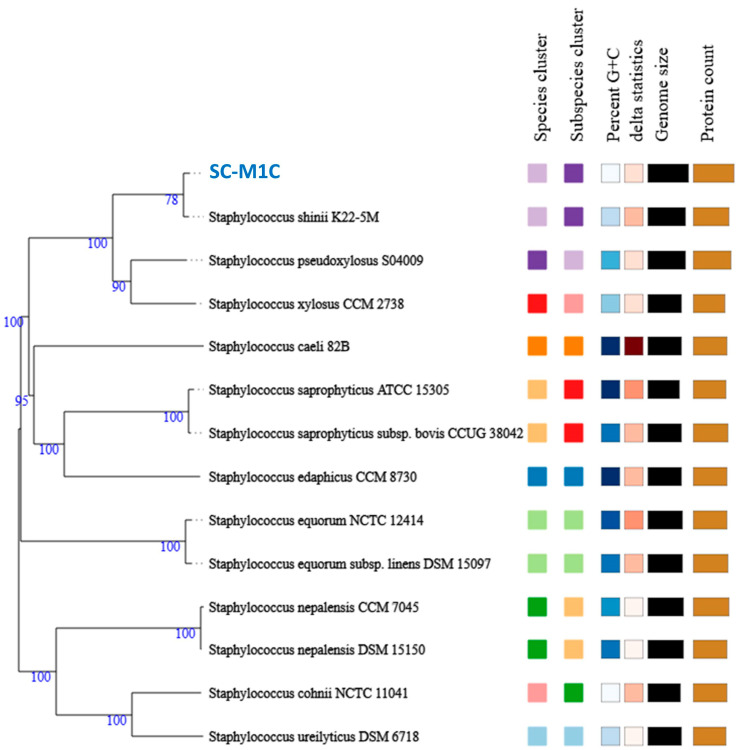
A phylogenomic tree constructed by the Type Strain Genome Server (TYGS). The tree is based on the genome of *S. shinii* strain SC-M1C and its top-related type strains. Confidence values are displayed near the nodes. The taxonomic status of these type strains were checked and corrected according to the List of Prokaryotic names with Standing in Nomenclature.

**Figure 4 microorganisms-13-01866-f004:**
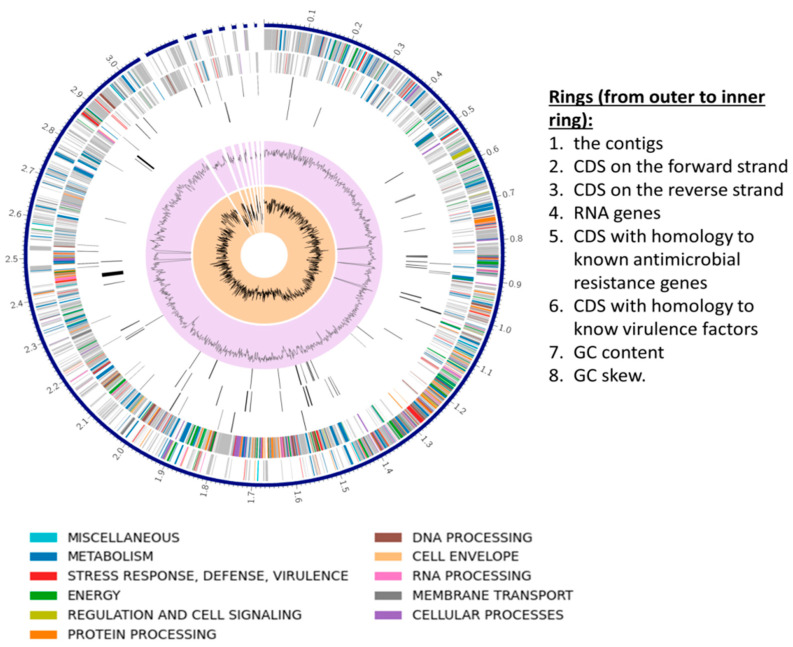
A circular diagram represents the genome map of *S. shinii* strain SC-M1C.

**Figure 5 microorganisms-13-01866-f005:**
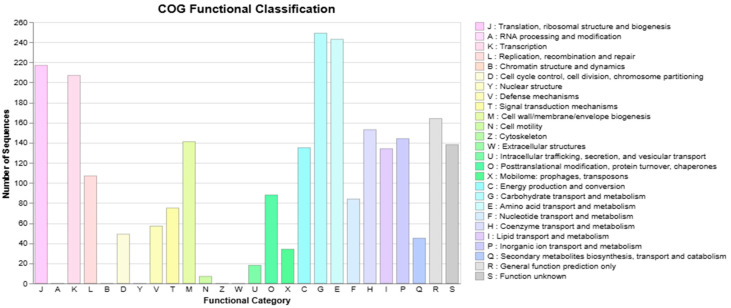
Clusters of orthologous genes (COG) functional classification of coding DNA sequences of the *S. shinii* strain SC-M1C genome.

**Figure 6 microorganisms-13-01866-f006:**
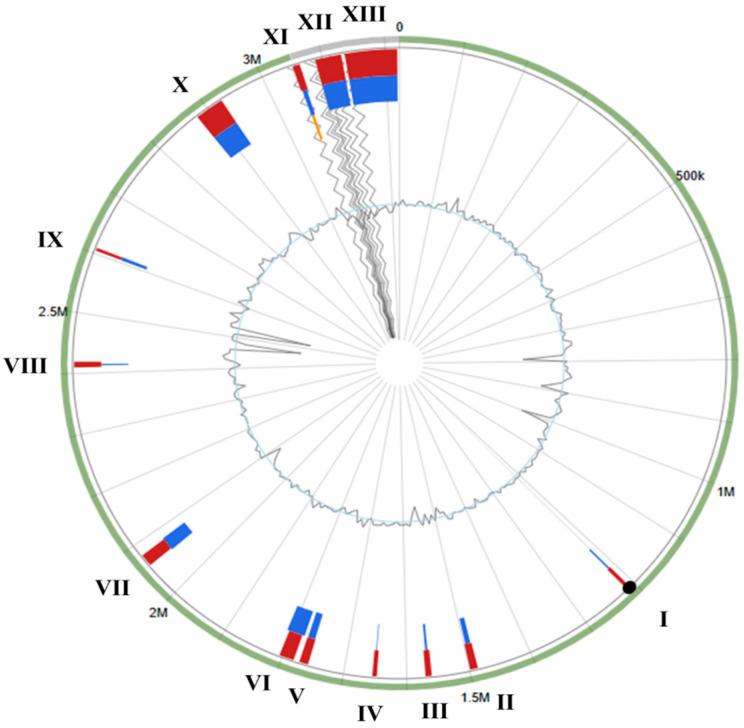
Genomic islands (GIs) identified by the IslandViewer 4 server within the *S. shinii* strain SC-M1C genome. The wavy lines represent the ends of contigs.

**Figure 7 microorganisms-13-01866-f007:**
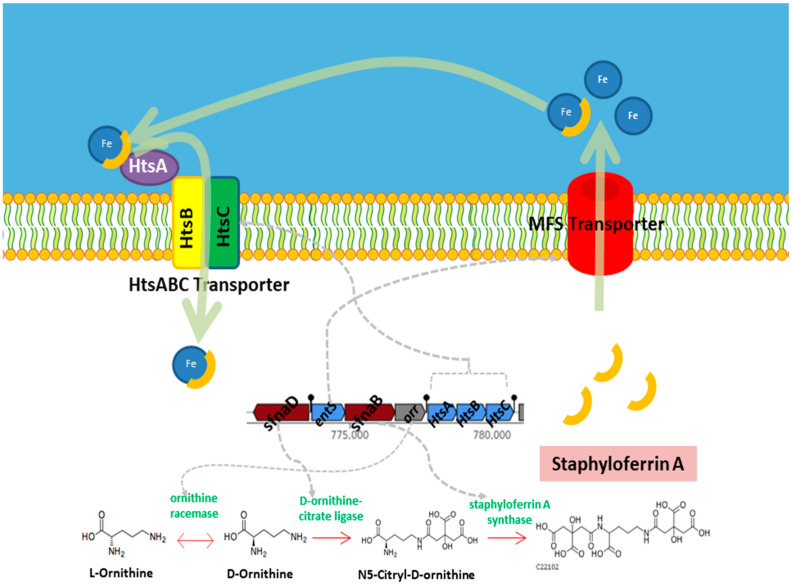
Proposed diagram of the production of staphyloferrin A in *S. shinii* strain SC-M1C genome.

**Figure 8 microorganisms-13-01866-f008:**
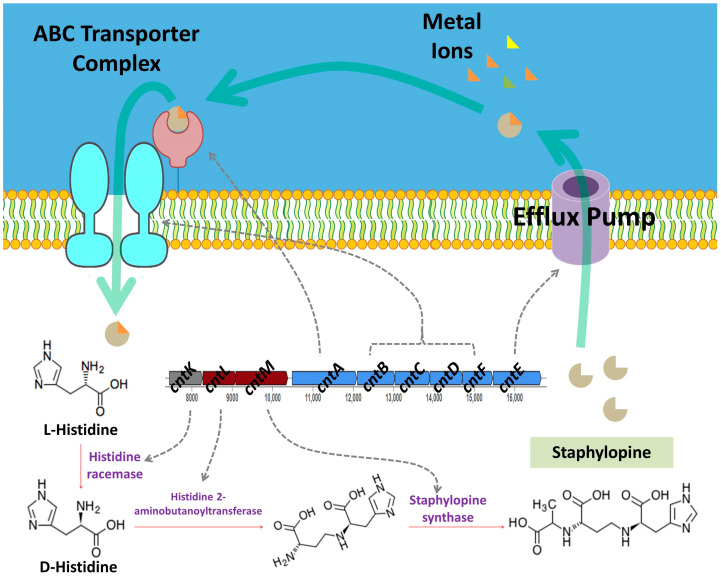
Proposed diagram of the production of staphylopine in the *S. shinii* strain SC-M1C genome.

**Table 1 microorganisms-13-01866-t001:** The genomic relatedness of the *S. shinii* strain SC-M1C to its closely related type strains. The table lists various genomic parameters, including digital DNA–DNA hybridization (dDDH), the difference in GC content (∆GC), and the average nucleotide identity (ANI) using BLAST-based alignment (ANIb) and MUMmer-based alignment (ANIm) with default parameters. The taxonomic statuses of these strains were checked and corrected according to the List of Prokaryotic Names with Standing in Nomenclature (https://lpsn.dsmz.de/, accessed on 24 May 2025).

Strain	NCBI Accession	ANIm (%)	ANIb (%)	dDDH (%)	GC (%)
*Staphylococcus shinii* K22-5M	GCF_017583065.1	99.40	99.09	94.8	0.15
*Staphylococcus pseudoxylosus* S04009	GCF_003697915.1	91.94	91.15	44.3	0.46
*Staphylococcus xylosus* CCM 2738	GCF_002732165.1	91.16	90.70	41.5	0.33
*Staphylococcus saprophyticus* CCUG 38042	GCF_002902545.1	85.04	80.71	24.8	0.65
*Staphylococcus saprophyticus* ATCC 15305	GCF_000010125.1	85.00	80.72	24.6	0.85
*Staphylococcus edaphicus* CCM 8730	GCF_002614725.1	84.76	80.34	24	0.92
*Staphylococcus caeli* 82B	GCF_900097965.1	84.79	79.70	23.7	0.95
*Staphylococcus equorum* NCTC 12414	GCF_003970515.1	84.60	79.77	23.7	0.77
*Staphylococcus equorum* subsp. linens DSM 15097	GCF_002901955.1	84.38	79.58	23.4	0.67
*Staphylococcus nepalensis* DSM 15150	GCF_002902745.1	84.81	78.57	23.3	0.63
*Staphylococcus nepalensis* CCM 7045	GCF_014635045.1	84.70	78.40	22.9	0.59
*Staphylococcus ureilyticus* DSM 6718	GCF_002902235.1	84.41	78.12	22.6	0.16
*Staphylococcus cohnii* NCTC 11041	GCF_002902365.1	84.22	78.32	22.4	0.1

**Table 2 microorganisms-13-01866-t002:** A summary of RAST and Prokka annotations of *S. shinii* strain SC-M1C genome.

	RAST	Prokka
Total CDs	3077	3032
CDs with functional assignment	2285	1946
Hypothetical CDs	792	1086
rRNA	2	1
tRNA	42	38

**Table 3 microorganisms-13-01866-t003:** Predicted virulence factors identified in the *S. shinii* SC-M1C genome using the virulence factor database (VFDB).

Virulence Class	Virulence Factor	Associated Gene(s)
Adherence	Autolysin; elastin-binding protein; fibronectin-binding proteins	*atl*; *ebp*; *fnbA*
Enzyme	Lipase; serine V8 protease; thermonuclease	*geh*; *sspA*; *nuc*
Immune evasion	Capsule; polyglutamic acid capsule (Bacillus); polysaccharide capsule (Bacillus)	*—*; *capB*, *capC*; *galE*
Iron uptake	Periplasmic-binding protein-dependent ABC transport system (Vibrio-like)	*vctC*
Nutritional factor	Allantoin utilization (Klebsiella-like)	—

**Table 4 microorganisms-13-01866-t004:** Antimicrobial resistance (AMR) genes identified in the genome of *S. shinii* strain SC-M1C, associated mechanisms, and corresponding antibiotic classes.

AMR Mechanism	Gene	Antibiotic Class
Antibiotic inactivation enzyme	*BlaZ* family	β-lactams
	*FosB*	Fosfomycin
	*Mph*(C) family	Macrolides
Antibiotic resistance gene cluster, cassette, or operon	*TcaB*, *TcaB2*, *TcaR*	Glycopeptides
Antibiotic target modification or protection	*Alr*	Cell wall synthesis inhibitors
Antibiotic target modification or protection	*Ddl*	Cell wall synthesis inhibitors
	*EF-G*, *EF-Tu*	Protein synthesis inhibitors
	*folA*, *Dfr*, *folP*	Antifolates
	*gyrA*, *gyrB*	Fluoroquinolones
	*inhA*, *fabI*	Fatty acid synthesis inhibitors
	*Iso-tRNA*	Protein synthesis inhibitors
	*kasA*	Fatty acid synthesis inhibitors
	*MurA*	Cell wall synthesis inhibitors
	*Rho*	Transcription inhibitors
	*rpoB*, *rpoC*	Rifamycins
	*S10p*, *S12p*	Aminoglycosides
Efflux pumps conferring antibiotic resistance	*NorA*	Fluoroquinolones
	*YkkCD*	Multidrug efflux
Gene conferring resistance via absence	*gidB*	Aminoglycosides (loss of methylation)
Protein-altering cell wall charge	*GdpD*, *MprF*, *PgsA*	Cationic antimicrobial peptides
Regulator modulating antibiotic resistance gene expression	*BceR*, *BceS*, *LiaF*, *LiaR*, *LiaS*	Glycopeptides and antimicrobial peptides

**Table 5 microorganisms-13-01866-t005:** Predicted biosynthetic gene clusters identified in the *S. shinii* SC-M1C genome using AntiSMASH version 6.0.

BGC	Type	Size (bp)	Most Similar Known Cluster	Identity
1	Opine-like metallophore, terpene	37,511	Staphylopine	100%
2	RiPP-like	10,281		
3	T3PKS	41,169		
4	NI siderophore	32,991	Staphyloferrin A	100%
5	Cyclic lactone autoinducer	20,709		
6	Terpene	20,818		
7	Terpene precursor	20,890		
8	Terpene precursor	20,953		
9	Lanthipeptide class i	24,084		
10	NRPS	56,796		
11	NRPS	44,376		
12	NRPS	44,328		

**Table 6 microorganisms-13-01866-t006:** Osmoregulation and salt tolerance genes identified in *S. shinii* strain SC-M1C. Genes include Na^+^/H^+^ antiporters and compatible solute transporters involved in maintaining osmotic balance in marine conditions.

Function	Genes	COG(s)	Role	References
Na^+^/H^+^ antiport system (multi-subunit)	*mnhA1*, *mnhB1*, *mnhC1*, *mnhD1*, *mnhE1*, *mnhF1*, *mnhG1*	COG0651, COG2111, COG1006, COG1863, COG2212, COG1320	Maintains intracellular pH and osmotic balance under high-salinity conditions.	[40]
Na^+^/H^+^ antiporter	*nhaC*	COG1757	Regulates pH and ion homeostasis in saline environments.	[40,41]
Compatible solute uptake	*opuD*	COG1292	Imports glycine betaine to protect against osmotic stress.	[40,42]

**Table 7 microorganisms-13-01866-t007:** Genes related to nutrient acquisition identified in *S. shinii* SC-M1C genome and their potential roles.

Nutrient	Genes	COG(s)	Role	References
Methionine	*metP*, *metN2*	COG2011, –	Active transport of methionine in nutrient-poor marine habitats.	[43]
Nitrogen (nitrate/nitrite assimilation)	*nasD*	COG1251	Reduces nitrite to ammonium for nitrogen assimilation.	[44,45]
Phosphate	*pstS*	–	Scavenges inorganic phosphate in oligotrophic waters.	[46,47]
Thiamine	*ykoD*	COG1122	Imports thiamine, essential for metabolic processes.	[48,49]
Iron	*fhuD*	–	Facilitates acquisition of iron under limiting conditions.	[50,51]

**Table 8 microorganisms-13-01866-t008:** Stress response-related genes and their potential roles in the *S. shinii* SC-M1C genome.

Stress Type	Genes	COG(s)	Role	References
Oxidative stress and DNA repair	*nth*, *trxB*, *recA*	–, COG0492, –	Protects from oxidative damage and promotes DNA repair.	[52]
Heavy metal stress (copper)	*copA*, *copB*	–	Exports excess copper to prevent toxicity.	[53]
Detoxification and efflux	*sepA*, *mdtD*	–	Removes toxic or host-derived compounds.	[54,55]

**Table 9 microorganisms-13-01866-t009:** Genes associated with potential aromatic hydrocarbon degradation in *S. shinii* SC-M1C.

Genes	COG(s)	Role	References
*catE*	COG2514	Acts as a key catechol 2,3-dioxygenase, performing meta-cleavage of aromatic rings, a critical step in the breakdown of petroleum hydrocarbons.	[56]
*mhqR*, *mhqA*, *mhqO*, *mhqD*	COG1846, COG0346, COG0400	This *mhq* cluster enables the degradation of a variety of aromatic compounds.	[57]

## Data Availability

The original data presented in the study are available in NCBI under the following accessions: PRJNA1280784 (Bioproject), SAMN49534678 (BioSample), SRR34102925 (Sequence Read Archive), and GCF_051216755.1 (Assembled genome).

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
