# Peer review of "Genomic Characterization of Marine Staphylococcus shinii Strain SC-M1C: Potential Genetic Adaptations and Ecological Role"

_microorganisms, 2025, doi:10.3390/microorganisms13081866_

Round 1

Reviewer 1 Report

Comments and Suggestions for Authors

The manuscript titled "Genomic Characterization of the Marine Bacterium Staphylococcus shinii Strain SC-M1C: Insights into Genetic Adaptations and Ecological Roles in Marine Ecosystems" presents a well-organised and timely study focused on a previously unreported marine isolate of Staphylococcus shinii. However, it requires substantial revision to refine the interpretation of genomic findings, improve clarity and precision in data presentation, and temper unsupported ecological claims. Addressing these concerns will enhance the scientific robustness and readability of the manuscript.

Major Comments

  1. Many of the adaptive features identified (e.g., resistance genes, virulence factors, biosynthetic gene clusters) are described speculatively without comparative or functional validation, leading to overstated ecological significance.
  2. The Discussion should be revised to more carefully distinguish between gene presence and proven function.
  3. Furthermore, figures, tables, and supplementary materials need improved labelling and integration to support the narrative effectively

Minor comments

  1. Shorten the title for clarity and impact. Consider removing redundant phrases with more concise alternatives.
  2. Provide citation support for the statement in Line 49 that marine microbial adaptations challenge traditional views of microbial evolution. Include a recent reference that highlights this shift.
  3. Reword Line 51 for clarity. "Distinct seasonal variations" is vague—briefly state which seasonal factors (e.g., temperature, salinity) vary in the Red Sea.
  4. In Line 56, the phrase "making them key players" is repetitive. Combine with the earlier sentence or rephrase to avoid redundancy.
  5. Lines 60–62 should be rewritten for clarity. The sentence is too long and convoluted. Break it into two sentences: one describing well-studied marine bacteria, and one introducing Staphylococcus as underrepresented.
  6. In Line 65, do not generalize that several Staphylococcus species are pathogens. Clarify that the genus includes both pathogenic and non-pathogenic species and specify relevance to environmental isolates.
  7. The transition in Line 88 ("This study fills that gap…") is abrupt. Start with a sentence summarizing the goal of the study before stating its novelty.
  8. Lines 89–91 list genomic traits of adaptation. Consider citing examples from similar marine studies that report the same traits in other bacteria for comparison.
  9. Line 95 says “newly identified marine isolate,” but the species is not new. Use “marine isolate of S. shinii” instead to avoid confusion with species novelty.
  10. Proofread the entire Introduction for consistent use of tense. For instance, the switch between present and past tense in Lines 84–96 can be made more consistent by keeping the background in the past tense and study goals in the present tense.
  11. In Line 102, clarify whether sponge identification was confirmed morphologically or via molecular taxonomy (e.g., 18S rRNA or COI gene). Include species identification method if available.
  12. In Line 106, state whether seawater was autoclaved or 0.22 μm filtered for sterilization. The current phrase “filtered and sterilized” is unclear.
  13. In Line 107, clarify whether the 10 mL homogenization was performed mechanically (e.g., vortex, stomacher) or manually. Provide the device/method used.
  14. In Line 112, report how many isolates were recovered initially and what criteria were used to select SC-M1C for further analysis.
  15. In Line 120, specify the duration of culture incubation (“till growth” is ambiguous). Mention approximate OD or growth phase.
  16. In Line 140, clarify the JSpecies parameters used (e.g., ANIb or ANIm) and whether local BLAST or MUMmer alignment was employed.
  17. In Line 163, provide the incubation time used to observe colony morphology at 37 °C (24 h, 48 h, etc.). The timing matters for reproducibility.
  18. In table 1, make sure to fix the overlapping , and include the contigs number, genome completeness and contamination
  19. Figure 2b lacks a scale bar. Add a scale bar to the microscopic image to help readers interpret cell size.
  20. In Line 170, include the number of sequencing reads generated and post-filtering read statistics to support the N50 and contig metrics.
  21. In Table 6 (Line 294), describe whether these osmoregulation genes were present as single copies or gene families. Were they clustered?
  22. In Line 324, avoid repeating the Introduction. Summarize key findings briefly and focus more on interpretation rather than restating background.
  23. In Line 330, strengthen the interpretation by comparing SC-M1C's genome size, GC content, or key traits with other marine Staphylococcus strains or related genera.
  24. In Lines 339–341, the general definition of genomic islands can be reduced or moved to the Introduction. Instead, focus on how the identified GIs in SC-M1C contribute specifically to marine survival.
  25. In Lines 349–351, relate GI gene functions more directly to marine stressors—e.g., high salt, limited nutrients, or oxidative conditions—rather than broad terms like "challenging conditions."
  26. End the section with a clear, concise conclusion paragraph highlighting the novel insights provided by the study and suggesting specific future research directions (e.g., transcriptomic validation, bioactivity screening).

Author Response

  1. Authors responses to the reviewers’ comments.

    We are grateful to the reviewers for their constructive feedback, which has significantly improved the manuscript. Here are our detailed responses to the reviewers’ comments:

    Reviewer 1

    1. Comment: Many of the adaptive features identified (e.g., resistance genes, virulence factors, biosynthetic gene clusters) are described speculatively without comparative or functional validation, leading to overstated ecological significance.

    Response: Thanks a lot for highlighting the shortcomings. In response to this comment, we updated the text in the “Abstract” section, lines 23 to 27, and we denoted these functions as “potential” based on the reported purposes of these gene products in the literature.

    1. Comment: The Discussion should be revised to more carefully distinguish between gene presence and proven function.

    Response: We appreciate your advice concerning the uncertainty of the “Discussion” section regarding the distinction between gene presence and proven function. In response to that, we revised this section to make it clear between the presence of genes and their reported function (lines 313- 324, 348-359, 374-400).

    1. Comment: Furthermore, figures, tables, and supplementary materials need improved labelling and integration to support the narrative effectively

    Response: Thank you for highlighting the importance of aligning the narrative with the visual explanations in the figures and tables. All integrated figures and tables have been carefully revised and updated by your recommendations.

    1. Comment: Shorten the title for clarity and impact. Consider removing redundant phrases with more concise alternatives.

    Response: Thanks a lot for your recommendation. We revised and shortened the title and removed any irrelevant words as per your instructions.

    1. Comment: Provide citation support for the statement in Line 49 that marine microbial adaptations challenge traditional views of microbial evolution. Include a recent reference that highlights this shift.

    Response: We appreciate your viewpoint, and we added a relevant reference to this statement “reference number 8”.

    1. Comment: Reword Line 51 for clarity. "Distinct seasonal variations" is vague—briefly state which seasonal factors (e.g., temperature, salinity) vary in the Red Sea.

    Response: Thanks for your comment. In response to the reviewer’s helpful suggestion, we have revised the text to improve clarity.

    1. Comment: In Line 56, the phrase "making them key players" is repetitive. Combine with the earlier sentence or rephrase to avoid redundancy.

    Response: Thanks a lot for highlighting this. We agree with the reviewer’s observation and have amended the relevant Line accordingly.

    1. Comment: Lines 60–62 should be rewritten for clarity. The sentence is too long and convoluted. Break it into two sentences: one describing well-studied marine bacteria, and one introducing Staphylococcus as underrepresented.

    Response: Thanks a lot for highlighting this. We agree with the reviewer’s observation and have amended the relevant Lines accordingly.

    1. Comment: In Line 65, do not generalize that several Staphylococcus species are pathogens. Clarify that the genus includes both pathogenic and non-pathogenic species and specify relevance to environmental isolates.

    Response: We appreciate the reviewer’s insightful comment and have incorporated the recommended change in the revised version (line 56- 64).

    1. Comment: The transition in Line 88 ("This study fills that gap…") is abrupt. Start with a sentence summarizing the goal of the study before stating its novelty.

    Response: We appreciate your point of view. We have revised the section to better reflect the reviewer’s comment, while maintaining alignment with our study scope (line 79).

    1. Comment: Lines 89–91 list genomic traits of adaptation. Consider citing examples from similar marine studies that report the same traits in other bacteria for comparison.

    Response: Thanks for your comment. We appreciate the reviewer’s insightful comment and have incorporated a few studies that reported comparable bacterial traits

    1. Comment: Line 95 says “newly identified marine isolate,” but the species is not new. Use “marine isolate of S. shinii” instead to avoid confusion with species novelty.

    Response: Thanks for reporting that for us. The line has been revised and updated accordingly.

    1. Comment: Proofread the entire Introduction for consistent use of tense. For instance, the switch between present and past tense in Lines 84–96 can be made more consistent by keeping the background in the past tense and study goals in the present tense.

    Response: Thanks for reporting that for us. The “Introduction” section has been revised, and tenses have been updated accordingly.

    1. Comment: In Line 102, clarify whether sponge identification was confirmed morphologically or via molecular taxonomy (e.g., 18S rRNA or COI gene). Include species identification method if available.

    Response: Thanks for your comment. The method of sponge identification has been stated and mentioned in Lines 92- 94.

    1. Comment: In Line 106, state whether seawater was autoclaved or 0.22 μm filtered for sterilization. The current phrase “filtered and sterilized” is unclear.

    Response: We appreciate your concerns. The method has been clarified to indicate that the seawater was filtered from impurities, then autoclaved for sterilization (line 97).

    1. Comment: In Line 107, clarify whether the 10 mL homogenization was performed mechanically (e.g., vortex, stomacher) or manually. Provide the device/method used.

    Response: Thanks for reporting that for us. The line has been revised and updated to state clearly the mechanical method of homogenization (Line 100).

    1. Comment: In Line 112, report how many isolates were recovered initially and what criteria were used to select SC-M1C for further analysis.

    Response: Thanks a lot for highlighting this point to us. We updated the corresponding lines to clarify the number and criteria used in SC-M1C selection (Lines 101 – 104).

    1. Comment: In Line 120, specify the duration of culture incubation (“till growth” is ambiguous). Mention approximate OD or growth phase.

    Response: We appreciate your concerns regarding the accuracy of growth conditions. These conditions have been revised and clarified in Line 103.

    19: Comment: In Line 140, clarify the JSpecies parameters used (e.g., ANIb or ANIm) and whether local BLAST or MUMmer alignment was employed.

    Response: Thanks for your comment. The used parameters have been revised and stated clearly in Lines 129- 132.

    1. Comment: In Line 163, provide the incubation time used to observe colony morphology at 37 °C (24 h, 48 h, etc.). The timing matters for reproducibility.

    Response: Thanks a lot for highlighting that for us. We appreciate the reviewer’s insightful comment and have incorporated the incubation conditions in Line 104.

    1. Comment: In table 1, make sure to fix the overlapping, and include the contigs number, genome completeness, and contamination.

    Response: Thanks a lot for your comment. The data has been double-checked and fixed

    accordingly using the provided NCBI Accession numbers.

    1. Comment: Figure 2b lacks a scale bar. Add a scale bar to the microscopic image to help readers interpret cell size.

    Response: Thanks this important comment. We add the appropriate scale to the figure.

    23: Comment: In Line 170, include the number of sequencing reads generated and post-filtering read statistics to support the N50 and contig metrics.

    Response: Thanks a lot, we appreciate this comment. Lines 159 and 160 have been revised and the recommended data have been added.

    1. Comment: In Table 6 (Line 294), describe whether these osmoregulation genes were present as single copies or gene families. Were they clustered?

    Response: Thanks a lot for your comment. The paragraph in Line 270- 274 has been edited and updated to include the required clarifications.

    1. Comment: In Line 324, avoid repeating the Introduction. Summarize key findings briefly and focus more on interpretation rather than restating the background.

    Response: Thanks a lot for your valuable comment. The paragraph has been updated in Lines 315- 326 to avoid any repetition and to emphasis more on the data analysis.

    1. Comment: In Line 330, strengthen the interpretation by comparing SC-M1C's genome size, GC content, or key traits with other marine Staphylococcus strains or related genera.

    Response: We appreciate your comment. We rewrote the paragraph to include the missing point as per your recommendations in Line 323 to 326.

    1. Comment: In Lines 339–341, the general definition of genomic islands can be reduced or moved to the Introduction. Instead, focus on how the identified GIs in SC-M1C contribute specifically to marine survival.

    Responses: Thanks a lot for your suggestion. We have moved part of the paragraph to the introduction section (lines 327 to 336), and edited part of it to match your recommendations

    28: Comment: In Lines 349–351, relate GI gene functions more directly to marine stressors—e.g., high salt, limited nutrients, or oxidative conditions—rather than broad terms like "challenging conditions."

    Response: Thanks for your valuable comment. We updated this paragraph as per your recommendations, new updates are in Line 331- 333.

    1. Comment: End the section with a clear, concise conclusion paragraph highlighting the novel insights provided by the study and suggesting specific future research directions (e.g., transcriptomic validation, bioactivity screening).

    Response: We appreciate your comment. We revised and updated the discussion section and made the required changes.

Reviewer 2 Report

Comments and Suggestions for Authors

Review of the manuscript entitled “Genomic Characterization of the Marine Bacterium Staphylococcus shinii Strain SC-M1C: Insights into Genetic Adaptations and Ecological Roles in Marine Ecosystems”, by El Samak et al., submitted for consideration to microorganisms as a Research Article.

General comment: This manuscript presents a genomic analysis of a Staphylococcus shinii strain isolated from the marine environment, specifically from a sea sponge in the Red Sea. In general, the authors combine bioinformatic tools to propose mechanisms of environmental adaptation of this understudied microbial taxon to the marine environment. The paper is written quite well, but I have some important concerns regarding the experimental design and the methodology used for microbial isolation. I also list some minor suggestions, potentially useful to improve tha manuscript once the main concerns are possibly solved.

Specific comments:

Title: I suggest revising the title to include the word "potential" before “genetic adaptations and ecological roles,” as the current phrasing implies a confirmed ecological function and adaptation, while the findings are largely based only on genomic predictions. A more accurate title might be:
“Potential Genetic Adaptations and Ecological Roles of a novel bacterium (Staphylococcus shinii Strain SC-M1C) isolated from a Red Sea sponge”. This revision would better reflect the preliminary nature of the results and maintain scientific caution regarding functional inferences and bacterial origin drawn from genomic data alone.

Abstract: I recommend revising the abstract to use more cautious language when referring to genetic adaptation and ecological roles, as the current wording implies experimentally confirmed findings. Terms like “potential” would more accurately reflect the inferential nature of genomic analysis.

Abstract and Introduction: The major issue the Authors need to solve is the possible origin of Staphylococci in the sea. Are these terrestrial contaminants or actual marine bacteria? Staphylococcus species are not typically considered marine bacteria. They are ubiquitous most commonly found on skin and mucous membranes of humans and animals, as well as in soil, water, and food. I suggest the authors strengthen the introduction by including additional references on the presence and roles of Staphylococcus species in the marine environments. For instance, in line 70, the Authors cite 2 papers regarding isolation of Staphylococcus areus from marine environment, but one of them (ref 11), actually isolated this species from a freshwater environment. A more robust literature context is needed to support the novelty and relevance of isolating S. shinii from a marine sponge in this study.

M&M: I have a major concern regarding the isolation method used in this study. The authors employed R2A agar supplemented with 2% NaCl, which does not adequately replicate the natural salinity of the Red Sea, typically around 3.8% (almost double). This raises doubts about whether the isolated S. shinii strain is truly a marine (or, marine-adapted) bacterium or rather a halotolerant terrestrial strain capable of surviving for some time in (mildly saline, 2% NaCl) marine environment. I also argue that the lack of controls in the culturing procedure (media without sponge inoculum) raises the issue that the strain obtained might just be a lab contaminant. To support their claim of a marine origin, the authors should provide more information about the salinity at the sampling site and clarify whether the culture conditions selected were appropriate for isolating an obligate/purely marine bacterium. Also, the authors should demonstrate its ability to grow on standard marine media (such as Marine Broth) and test salinity and nutrient contents that more frankly reflect those of the natural environment of origin.

Line 105: please introduce what the abbreviation ATM means.

Line 112: you state that microorganisms were incubated at 30°C for a week, but later in line 166 you state that the colonies formed at 37°C. Please indicate the right temperature for your experiment.

Results/Discussion: These sections are in general well-structured and rich in data, however, there are several parts with overstatements. Many sentences imply confirmed functional roles (e.g., line 338: “S. shinii contributes to microplastic degradation”) based solely on gene annotations. Or line 230: “These results exhibit a completely new module for multidrug-resistance studies.”

Also, in the discussion, there is a limited comparison with other known marine staphylococci or sponge-associated microbiota. Please add a comparison of these findings with published genomes/metagenomes from other sponge-associated bacteria (especially checking for Staphylococcus spp. presence in published sponge microbial metagenomes).

Hints for further insights into the isolated strain: While the authors highlight genomic features associated with heavy metal resistance, I recommend they also examine whether the genome harbors genes related to detoxification of heavy metals, as well as degradation of common marine organic pollutants such as petroleum hydrocarbons. You could refer to some recent papers to broadening  your discussion (Dell’anno et al., 2021, 2023). Once the abovementioned major comments are solved, extending the possible relevance of this strain beyond ecological roles, by checking for potential implications in bioremediation, would increase the significance of this paper.

[Dell’Anno, F., et al (2023). Microbiome enrichment from contaminated marine sediments unveils novel bacterial strains for petroleum hydrocarbon and heavy metal bioremediation. Environmental Pollution, 317, 120772.]

[Dell’Anno, F., et al (2021). Bacteria, fungi and microalgae for the bioremediation of marine sediments contaminated by petroleum hydrocarbons in the omics era. Microorganisms, 9(8), 1695.]

Author Response

Authors responses to the reviewers’ comments.

We are grateful to the reviewers for their constructive feedback, which has significantly improved the manuscript. Here are our detailed responses to the reviewers’ comments:

Reviewer 2

  1. Comment: Title: I suggest revising the title to include the word "potential" before “genetic adaptations and ecological roles,” as the current phrasing implies a confirmed ecological function and adaptation, while the findings are largely based only on genomic predictions. A more accurate title might be: “Potential Genetic Adaptations and Ecological Roles of a novel bacterium (Staphylococcus shinii Strain SC-M1C) isolated from a Red Sea sponge”. This revision would better reflect the preliminary nature of the results and maintain scientific caution regarding functional inferences and bacterial origin drawn from genomic data alone.

Response: We gratefully appreciate the reviewer’s point of view and the newly suggested title. The manuscript’s title has been revised accordingly and the required changes have been made based on the recommendations from both reviewers.

  1. Comment: Abstract: I recommend revising the abstract to use more cautious language when referring to genetic adaptation and ecological roles, as the current wording implies experimentally confirmed findings. Terms like “potential” would more accurately reflect the inferential nature of genomic analysis.

Response: Thanks a lot for the insightful comment. We agree with the reviewer’s observation and have amended the Abstract section accordingly.

  1. Comment: Abstract and Introduction: The major issue the Authors need to solve is the possible origin of Staphylococci in the sea. Are these terrestrial contaminants or actual marine bacteria? Staphylococcus species are not typically considered marine bacteria. They are ubiquitous most commonly found on skin and mucous membranes of humans and animals, as well as in soil, water, and food. I suggest the authors strengthen the introduction by including additional references on the presence and roles of Staphylococcus species in the marine environments. For instance, in line 70, the Authors cite 2 papers regarding isolation of Staphylococcus aureus from marine environment, but one of them (ref 11), actually isolated this species from a freshwater environment. A more robust literature context is needed to support the novelty and relevance of isolating S. shinii from a marine sponge in this study.

Response: Thanks a lot for this in-depth analysis and recommendations. We appreciate the reviewer’s concerns, and in response to these, we have added references from 12 to 17 ( lines 56 to 64)

  1. Comment: M&M: I have a major concern regarding the isolation method used in this study. The authors employed R2A agar supplemented with 2% NaCl, which does not adequately replicate the natural salinity of the Red Sea, typically around 3.8% (almost double). This raises doubts about whether the isolated S. shinii strain is truly a marine (or, marine-adapted) bacterium or rather a halotolerant terrestrial strain capable of surviving for some time in (mildly saline, 2% NaCl) marine environment. I also argue that the lack of controls in the culturing procedure (media without sponge inoculum) raises the issue that the strain obtained might just be a lab contaminant. To support their claim of a marine origin, the authors should provide more information about the salinity at the sampling site and clarify whether the culture conditions selected were appropriate for isolating an obligate/purely marine bacterium. Also, the authors should demonstrate its ability to grow on standard marine media (such as Marine Broth) and test salinity and nutrient contents that more frankly reflect those of the natural environment of origin.

Response: Thanks a lot for this in-depth discussion. The strain SC-M1C was isolated 2022 as a part of previous metagenomic study (6). For the isolation of the twenty strains of the sponge, Marine agar was used for the isolation which was prepared from Sterile NSW. Also there were negative control in the culturing procedure. After strains purification, I was stored as -80 glycerol stock isolate. At the time refreshment of this isolates, R2A media was used for the its cultivation as mentioned in the methods section.  

Line 105: please introduce what the abbreviation ATM means.

Response: Done

Line 112: you state that microorganisms were incubated at 30°C for a week, but later in line 166 you state that the colonies formed at 37°C. Please indicate the right temperature for your experiment.

Response: We appreciate your valuable critical comments. The incubation temperature is 30â—¦C, which is corrected in (line 102, line 110)

  1. Comment: Results/Discussion: These sections are in general well-structured and rich in data, however, there are several parts with overstatements. Many sentences imply confirmed functional roles (e.g., line 338: “S. shinii contributes to microplastic degradation”) based solely on gene annotations. Or line 230: “These results exhibit a completely new module for multidrug-resistance studies.” Also, in the discussion, there is a limited comparison with other known marine staphylococci or sponge-associated microbiota. Please add a comparison of these findings with published genomes/metagenomes from other sponge-associated bacteria (especially checking for Staphylococcus spp. presence in published sponge microbial metagenomes).

Response: Thank you so much for insightful comment. We have revised the Results and Discussion sections to better reflect the reviewer’s comment, while maintaining alignment with our study scope. The modified sections highlighted with yellow color.

  1. Comment: Hints for further insights into the isolated strain: While the authors highlight genomic features associated with heavy metal resistance, I recommend they also examine whether the genome harbors genes related to detoxification of heavy metals, as well as degradation of common marine organic pollutants such as petroleum hydrocarbons. You could refer to some recent papers to broadening your discussion (Dell’anno et al., 2021, 2023). Once the abovementioned major comments are solved, extending the possible relevance of this strain beyond ecological roles, by checking for potential implications in bioremediation, would increase the significance of this paper.

[Dell’Anno, F., et al (2023). Microbiome enrichment from contaminated marine sediments unveils novel bacterial strains for petroleum hydrocarbon and heavy metal bioremediation. Environmental Pollution, 317, 120772.]

[Dell’Anno, F., et al (2021). Bacteria, fungi and microalgae for the bioremediation of marine sediments contaminated by petroleum hydrocarbons in the omics era. Microorganisms, 9(8), 1695.].

Response: Thanks a lot for your comment. We acknowledge the reviewer’s perspective; however, The analysis have been made and results have been analyzed in the discussion section (line 384 to 394) References from 79 to 82 have been added.
